# *Ut sophistes pictor*: An Introduction to the Sophistic Contribution to Aesthetics

## Clare Lapraik Guest

School of Oriental and African Studies, University of London, London WC1E 7HU, UK; 676048@soas.ac.uk

**Abstract:** This essay provides an introduction to the question of the contribution of the ancient sophists to aesthetics in Western art. It commences by examining the persistent analogies to visual arts in negative and positive discussions of sophistry, both philosophical and rhetorical, and proceeds to examine sophistic rhetoric in Gorgias, Aristides, Lucian, Philostratus and Byzantine *ekphrasis*, culminating with Philostratus' discussions of mimesis and *phantasia* in *Apollonius of Tyana*. The discussions of the relation of being and nonbeing in Gorgias' *On Nonbeing* and in Plato's *Sophist* form the ontological core of sophistic claims about imaginative invention and the sophistic advancement of voluntary illusion (*apatē*) as a means to poetic "justice" or "truth". Such claims should be considered in the light of the epistemological and ontological skepticism propounded by Gorgias. Although the opprobrium attached to sophistry obscures its later influence, we can nevertheless discern a sophistic aesthetic tradition focused on the reflective reception of artworks that re-emerges in the Renaissance. In the last section, I adumbrate the lines of study for examining a sophistic Renaissance in the visual arts, with attention to antiquarianism as an area where the significance of the beholder's imaginative projection suggests the endurance—or revitalization—of sophistic aesthetics.

**Keywords:** sophistry; aesthetics; Reception of Classical Antiquity; Philostratus; Gorgias; epideictic rhetoric; *Kairos*; pictorial illusion; imagination and *phantasia*; Plato's *Sophist*

## 1. Introduction

This essay considers the persistent allusions to painting and sculpture that recur in discussions of sophistry—in writings of the sophists themselves, in critical reflection on them, as in rhetorical treatises, and in philosophical and literary censure, most famously by Plato. It argues that accounts of art produced by or concerning sophists bequeath to European aesthetics the insight that the reception of artworks is not merely passive but reactive and recreative, and that it also illustrates the subjectivity of sensorial apprehension that lies at the base of cognition. Art becomes an object of philosophical discussion via analogies with the sophists' manipulation of appearances as critiqued by Plato and Aristotle and through the sophists' own celebration of the persuasive potentialities of *logos*. This also means that theoretical reflection on art is intermixed ab initio with analogies between verbal and visual arts. The affective capacity of speech that the sophists cultivated through stylistic means introduces a further element, namely, the audience whose responsiveness, malleability and receptive imagination was the target of persuasive speech. Even a rhetorical theorist who was not a sophist, such as Quintilian, is very sensitive to the orator's appeal to the *phantasia* of the audience in the exercise of responsive imagination (*Institutio oratoria* VI.2.27–31, 36). We might also see this essay as a contribution (or one stage of a contribution) towards the huge theme of the *phantasia* in relation to sophistic speech (see Bundy 1927; Webb 2009); here we start to consider the determining and enduring effects of that theme on Western art.

The enduring opprobrium of the term "sophist" served to obscure the sophistic inheritance in subsequent European art and aesthetics. In Renaissance treatises, we find it used pejoratively of such diverse phenomena as late Scholastic logic (Vasoli 1968, pp. 16, 43) and painterly illusion cultivated through the non-substantial qualities of color and shading, as

in the *paragone* debates initiated by Benedetto Varchi in 1546 (Guest 2016, pp. 303, 307–8).[1] Classical diatribes against the sophists were repeated by Marsilio Ficino, the translator of the Platonic corpus, as may be seen in his *Argumentum in Protagoram* (Ficino 1542, pp. 224–26), in which Plato as a healer of souls is contrasted with the sophists as "*malefici et venefici*", whose teaching is like a false siren song or the Circean enchantments ("*fallaces Sirenum cantus et noxia pocula Circes*"), although Protagoras is acknowledged to have transmitted ancient "theology" in his myth of Prometheus and Epimetheus.

The Italian Renaissance did, however, see the recovery of antique Greek literature, in the popularity of authors associated with the "Second Sophistic" such as Lucian and Philostratus (Philostratus 1961),[2] or the influence of late Byzantine *literari* such as Manuel Chrysoloras. The period also saw the flourishing of a range of literary modes with sophistic associations, including miscellaneous literature, forms concerned with occasion or *kairos* (e.g., Erasmus' *Adagia* or Montaigne's *Essais*), paradoxical encomia and *antilogoi*, *ekphrasis* and discussions of the imagination (Trinkaus 1983; MacPhail 2011; Katinis 2019; see Katinis in this special issue).[3] While apologists for sophistic were rare (an exception being Sperone Speroni (see Katinis 2018)), discussions pertaining to sophistry and Plato's *Sophist* also surface in Renaissance poetics, in, for example, Guido Mazzoni, Gregorio Comanini, Torquato Tasso and Francesco Patrizi (Plato 1993; Katinis 2018, pp. 106–36; Guest 2019, pp. 178–79). The tension between *kairos* and the topics as the systematic ordering of preformulated arguments or figures to be deployed on apposite occasions may be regarded as characteristic of many artistic forms of the Renaissance, such as emblems or devices and the *grottesche* (discussed below).[4] Alongside rediscovered Greek literature and philosophy, patristic writings also provided material on the sophists, with a work such as Clement of Alexandria's *Strōmateis* both rehearsing the diatribes of classical authors and exemplifying the kind of eclectic encyclopedism associated with sophistic *poikilia* or cultivation of variety.[5]

Although the influence of the sophists on Renaissance literature is increasingly acknowledged, the inheritance they imparted to aesthetics and art criticism has received less attention, despite the conspicuous presence of visual allusions and analogies in sophistic literature, starting with Gorgias. This theme could be developed following Trinkaus' recognition of the similarity between the Greek sophist and Renaissance humanist movements (Trinkaus 1983) or the perceptive observations by Nancy Struever, who notes the importance of aesthetic communication in humanism as a part of its sophistic inheritance and ponders whether the "key tenets of Humanistic rhetoric are analogous to those of Gorgias" (Struever 1970, p. 46 ff.). Our review of the artistic-sophistic allusions will be retrospective and conditioned by the reappearance and centrality to Renaissance artistic culture of the very topics privileged for analogies with sophistic in antique literature: perspective, illusion and the role of fantasy in artistic invention. The role of fantasy in the invention and reception of artworks is particularly notable in Renaissance antiquarianism, and scholars such as Leonard Barkan have noted the openness of the landscape of ruins, which admits the historical imagination as a collaborative force. Barkan remarks that the epigrams composed for recovered classical statues in the Belvedere garden assisted viewers in "composing . . . their own experience of beholding" without, however, linking these modes of reception to the sophistic inheritance (Barkan 1999, p. 207; Barkan 1993, pp. 138, 143). Forty years ago, John Onians noted the descriptive character of writings on art from Pliny to Procopius and their attention to the viewer's experience and imagination, so that the very stone and metal of sculpture seem to transform as they are beheld (Onians 1980). Onians discusses sophistic *encomia* in the Roman Empire but without relating them to the arguments of the earlier sophists. In short, the connection between sophistics and aesthetics is a topic that seems hidden in plain sight.

I would suggest that the presence of such undeclared and under-explored sophistic elements should constitute a third strand to add to the more thoroughly studied topics of Platonism (Panofsky 1968) and the dominant influence of Aristotelian-Scholastic concepts on Renaissance naturalism (Summers 1987). We might see sophistry in the artistic context

as a kind of unspoken Other to Platonism that concerns the admixture of deceit in idealism or the anxiety that the revelatory character of art may be merely a matter of effect: an admixture we might see, for example, in the motif of idealized *ignudi* juxtaposed with empty masks or *larvae* in Michelangelo and his followers, reprised in illusionist key by Caracci in the Galleria Farnese (Figure 1).[6] If Platonism in art had a basis in Cicero's claim in *Orator* (2.83.10) that the artist invents forms from access to the idea or draws forth forms that are immanent in matter (De divinatione II.21.48, perhaps following Aristotle, Metaphysics Γ 1002a), sophistic invention, as we shall see, draws on forms made by chance, on the basis of apparent resemblances, which we might therefore relate to *kairos*.[7]

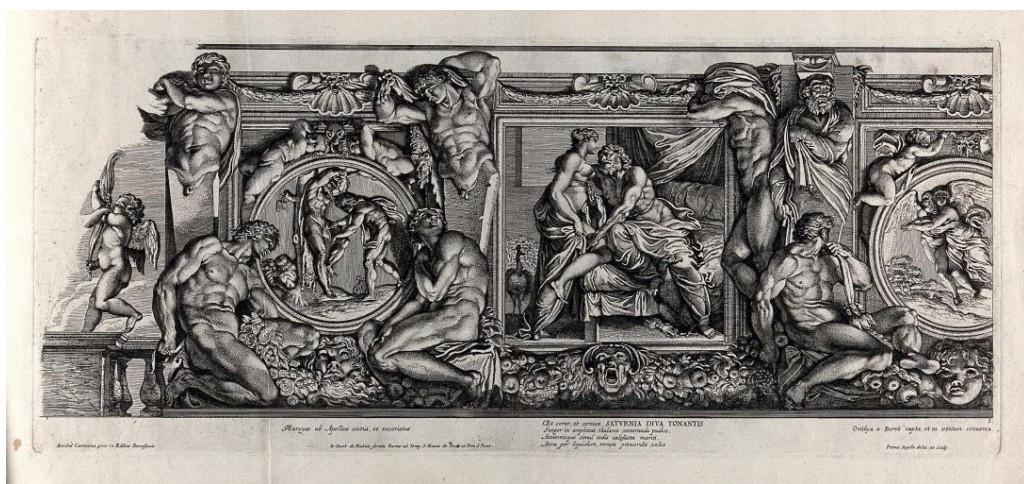

**Figure 1.** Pietro Aquila, etching from *Galeriae Farnesianae Icones* (c.1674) after Annibale Caracci, Farnese Gallery, Rome, 1597–1608. Marysyas flayed by Apollo, Juno and Jupiter and Oreithyia raped by Boreas, framed by *ignudi* and masks. Photo: Wellcome Collection, London.

The difference between the Platonic and sophistic accounts reflects different metaphysical presuppositions applied to a similar account of invention; when Cennini (1960, p. 1) speaks of the artist's *fantasia* as perceiving things "hidden under the shadow of natural objects," followed by an allusion to the artist's license to compose chimeric hybrids such as centaurs, we might see a conflation or muddling of sophistic themes with the metaphysical claims for artistic invention which would become so prominent in sixteenth century literature on Michelangelo.

The naturalistic tradition, advancing the heuristic role of perception, draws principally on Aristotelian conceptions of intellection and knowledge as founded on valid judgements concerning sensory experience and the role of the *phantasia* in judgement, action and conception–formation. From Aristotle's psychology, further developed by the Stoics, Neoplatonists, Augustine and the Scholastics, a formidable body of thought on perception, imagination and intellection was elaborated, in which sensory impressions exist in conformity with the objects of sensation and sense itself is understood as a kind of reasoning.[8] Thus, optics, which concerns both scientific laws and subjective distortions, accounts satisfactorily for viewpoint without recourse to sophistry—if we abstract from the persuasive intent of pictorial illusion and discount the imaginative activity of the viewer. From the naturalistic position, sophistic conceptions of art might appear to be intellectually redundant. Yet Federico Zuccari's treatise, *L'Idea de' pittori, scultori e architetti* (1607), the apogee of Aristotelian-scholastic art theory, is haunted by allusions to sophistry: he celebrates Proteus, archetype for the sophist, as the figure for painting (Zuccari 1768, pp. 95, 98).[9]

## 2. Transformative Reception

All of this prompts the question of how deeply the artistic analogies in ancient discussions of sophistry conditioned the experience or ideation of art and its perceived moral

intent. The range of such allusions is considerable, stretching from the analysis of reasoning in Aristotle's *Sophistical Refutations* 164b28 (Aristotle 1965) to ontological and epistemological issues in the Platonic dialogues to issues of genre, style and affect arising in rhetorical treatises, as when Cicero calls the sophists the source from which the *genus pictum* arose (*Orator* 96).[10] There are writings about art produced by sophists, such as the meditations in Philostratus' *Apollonius of Tyana*, or which were regarded as productions of sophistic rhetoric, such as the fictive pinacotheca in the *Imagines* of the Philostrati or Lucian's *De domo* (Philostratus the Elder et al. 1931; Lucian 1913).[11] Such writings contain suggestive explorations of the affective character of artistic illusion and the activity of the viewer.

A positive recognition of imaginative freedom and subjectivity in the response to art objects is one distinctive feature of sophistic meditations on art, as is argued in two significant passages in *Apollonius* (II.22, VI.19). The subjectivity of appearances is affirmed in skeptical philosophy (for example, in Sextus Empiricus' *Outlines of Pyrrhonism* [I.118–123]), but the affinity between sophistry and skepticism arises much earlier, with Gorgias' *On Nonbeing*, whose arguments, as we shall see, are reiterated by Philostratus.[12]

In sophistic discussions of art and aesthetic theory, artworks exist within a range of imaginary images or phantasms, encompassing the image in the artificer's mind and the reflections in the mind or memory of the beholder, which may in turn generate further potential artistic inventions. Artworks in this view exist in a kind of continuum of reflection that continuously engenders responses, a process signaled by the references to mirroring and the parallels between natural and human appearance-making that recur so often in sophistic allusions to art. Such parallels are set forth in the comparison of the productions of poets and painters to images seen in a moving mirror in Plato's *Republic* (596d–605c), at the opening of a condemnatory discussion of mimesis which culminates with the banishment of the poets, and in *Sophist* 266b–266d in a significant discussion of human and natural (divine) appearance-making, the former exemplified by painting and the latter including dreams, shadows and reflections (Plato 1993, 1997).

Plato's analogy between shadows or phantasms and human appearance-making recurs in rhetorical-sophistic form in Aristides' *Panathenaic Oration* (397) on *logos* as a mirror in which the soul gazes on the image of Athens (Aristides 1973), in the painting of Narcissus described in Philostratus' *Imagines* (I.23), and in Lucian's *De domo* 3 on the epideictic speech that reflects its subject like an echo, all discussed below.[13] In the *Panathenaic Oration* (157d), composed by an orator who claimed to experience oneiric revelations from Asclepius (recounted in his *Hieroi logoi*), the approach to Athens is likened to a joyful dream, so that the human appearance-making of epideictic *logos* is aligned with divine *phantasiai*.

In these cases, the reflective analogy does not connote verism but concerns the knowledge that arises from contemplation of phantasms, the representation of one image through another, or a painting through a speech and its mirroring in a shining surface or the beholder's mind. Such analogies are intensified in early Christian literature on church decoration, such as Prudentius' lines in *Peristephanon liber* (XII.39–42) on the painted decoration of the Vatican Baptistery that colors the waters beneath ("omnicolor vitreas pictura superne tinguit undas") while the reflected decoration seems to stir or dance in the stream ("credas moveri fluctibus lacunar"). In Prudentius' description, the reflection of the decoration is not an image of an image, at a further remove from the painted representation, which is itself derivative and phantasmal in an ontological hierarchy of the kind exemplified by Plato's fable of the cave in *Republic* (514a–517c). Rather, the reflection enlivens and, in a sense, perfects the painting, intimating that the painting is completed as it is reflected in the waters—and in the viewer's apprehension.

Reflective accounts of viewing images that thematize the process of seeing as much as the object seen culminate in Byzantine ekphrastic literature, where overwhelmed spectators are provoked to reflect on their own processes of vision, pushing beyond the subjectivity of imaginative reception into the subjectivity of optical experience. Space precludes extensive treatment, but amongst outstanding examples we might signal Procopius' meticulous account of the eye's pleasure in traversing the dome of Hagia Sophia (*On Buildings* I.1.47),

Photius' description of the spectator whirling around to view the "variegated spectacle" of the Palatine Chapel, who transfers his experience to the decoration that seems in ecstatic motion, or Michael of Thessalonika's description of the optical effects of the gold in Hagia Sophia, which seems to run and fuse with the moisture of the eyes themselves (Procopius 1940; Photius 1958, p. 187; Mango and Parker 1960, p. 237).[14] The engagement with the dynamic act of perception and the notion of matter in flux have repeatedly been noted in Byzantine architectural descriptions; we might see them as developing from a sophistic rhetorical tradition that foregrounded the optical and imaginative activity in apprehending an object.[15]

The material flux or material imagination so distinctive in Byzantine descriptions of striated marbles as "seas" or "flowery meadows" reflects a dynamic account of image reception that encompasses the fluid process of perception and the imaginative associations of an image.[16] (Figure 2) The reception of visual impressions (*phantasiai*) and the metaphoric associations (marbles as flowery meadows) are alike acts of the *phantasia*. In these emphatically phantastical accounts of the art of viewing artworks, the spectator is not merely inundated or overwhelmed: the reciprocal relationship between the psychagogic artwork and the reflective, transformative response of the viewer is highly elaborated.

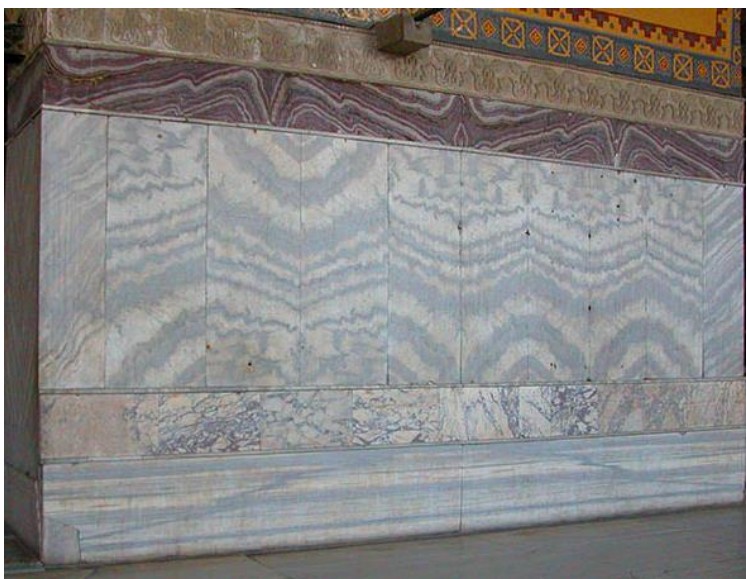

**Figure 2.** Hagia Sophia, Istanbul, West Gallery, detail of marble striation. Revetment showing (top-bottom) cipollino rosso, Proconesian, pavonazzetto and Proconnesian panels. Photo: Bob Atchison.

### 3. *Apatē*–Illusion

The Byzantine *ekphraseis* provide a kind of thematic *terminus ad quem* that illustrates the distinctive character of sophistic writings about art. The sophistic engagement with the subjectivity and malleability of perception and imaginative response should be regarded as fundamental to Western aesthetics and should not be overshadowed by the memorable presentation of the sophists of the classical period as the antagonists of the Platonic dialogues. As early as Gorgias, there is the recognition that the powerfully transformative effect of both *logos* and sight, repeatedly compared to enchantment or sorcery (*goēteia*) to signal the notion of an irresistible external agency, operates on the soul through voluntary participation in art's deception (*apatē*).[17] The pleasure-giving, sorrow-banishing power of enchantments are like transformative *goēteia* but are an error (*hamartia*) of the soul. Gorgias thus compares *logos* to a drug or *pharmakon* (*Helen* 14) that can poison or cure, a reference to either medicine or sorcery (or both). *Goēteia* would be an enduring metaphor for the psychagogic force of rhetorical literature. By the time of Aristotle's successor Theophrastus, psychagogic literature appears to have been distinguished from didactic prose (*didaskalia*), with the former regarded as entertainment yet likened to the leading of the soul in necro-

mancy.[18] Thus, in Aristides' *Panathenaic Oration*, a work that expands Gorgias' claims for *logos* through the celebration of Athenian *paideia*, we find a close repetition of *Helen* 10 and 14 on words as enchantments (*epōidai*) and *logos* as the finest drug or spell (*pharmakon*) the gods gave to humans.

Gorgias' *Helen*, the encomiastic speech that he teasingly proclaimed as a plaything or *jeu d'esprit* (*paignion*), provides a lucid examination of the beneficial or malefic effects of *logos* through its impact on the soul, exemplified by poetry that moves hearers to "shuddering fright and weeping pity and sorrowful longing" (*Helen* 9)—the tragic affects Aristotle would later discuss in the context of *katharsis* in *Poetics*. In a fragment quoted in Plutarch (*De gloria Atheniensium* 348c; *De audiendis poetis* 15d), Gorgias says that in tragedy, "He who deceives [*apatēsas*] is more just [*dikaioteros*] than he who does not deceive, and he who is deceived is wiser than he who is not."[19] The bewitching force ascribed to art requires the audience's imaginative participation in the work. Thus, in *De audiendis poetis* (15d), Plutarch prefaces Gorgias' comments on just or due deception in tragedy with a recollection of the sixth century BCE poet Simonides, who claimed that the Thessalians were too boorish to be beguiled by his poetry and thus remained refractory to *apatē*.[20] (Such obduracy contrasts with the repeated theme of animals that are awed or delighted by art and beauty, an exemplar of aesthetic compulsion rehearsed in Dio of Prusa's *Oration* XII and Lucian's *De domo*).

Thus, sophistic discussions of verbal and visual art, from the birth of rhetoric with Gorgias to the third-century Second Sophistic of Philostratus and the later rhetorical productions of the Byzantine era (a period now described by some scholars as the "Third Sophistic"), show sustained engagement with the relationship between aesthetic compulsion and aesthetic freedom: the compelling power of art to engage and move us and the imaginative subjectivity that characterizes our response.[21]

In contrast to these celebrations of the enchantments of sophistry and aesthetic experience, philosophical critiques of sophistry condemned its manipulation of appearances as fallacious arguments akin to the impressions and distortions of things seen from afar. Plato's *Sophist* (236c), for example, uses the conception of *mimēsis phantasikē* to describe large sculptures or paintings that adjust the proportions of the things they imitate for optical effect; what we might associate with later remains of the scenographic second style of Roman painting, as described by Vitruvius (*De architectura* VII.5.2) and codified by Auguste Mau (Figure 3). In *Republic* X (602d) *skiagraphia* (literally "shadow-painting"; three-dimensional modelling of images through placement of hues and tints) with its exploitation of optical distortions that bring "every confusion in our soul" and exploit the weakness in our nature, is condemned as a form of *goēteia*. Similarly, in *Sophistical Refutations* (164b28 and 165a22) Aristotle reiterates the comparison between apparent wisdom (*phainomenē sophia*) and appearances viewed from a distance.[22] Perspectivism therefore provides philosophers with the analogy for the sophist's presentation of false beliefs and relativism.

The theme of illusionistic depth also appears in a rhetorical, non-philosophical condemnation of over-embellishment after the manner of Gorgias in the critical essays of Dionysius of Halicarnassus (1974), in which the styles of Isocrates and Plato, criticized for their lapses into Gorgianic "vulgarity," are likened to objects fashioned in relief, *glyptos* and *toreutos* (*Demosthenes* 51). Dionysius' analogies are not just an effect of the metaphoric language used in discussion of the ornaments of speech, such as we find amply deployed in Cicero's *Orator*, but are instead focused on the *experience* of oratorical language, so that certain artistic styles illustrate the sensation or reflective encounter of oratory, with Gorgias' "dithyrambic" style forming a consistent antitype for overblown and frigid ornament.[23] Dionysius also uses music in such affective analogies: the experience of Isocrates' orations is like listening to libation-music on reed pipes or Dorian or enharmonic melodies, while Demosthenes' speeches are like the Corybantic dances (*Demosthenes* 22).

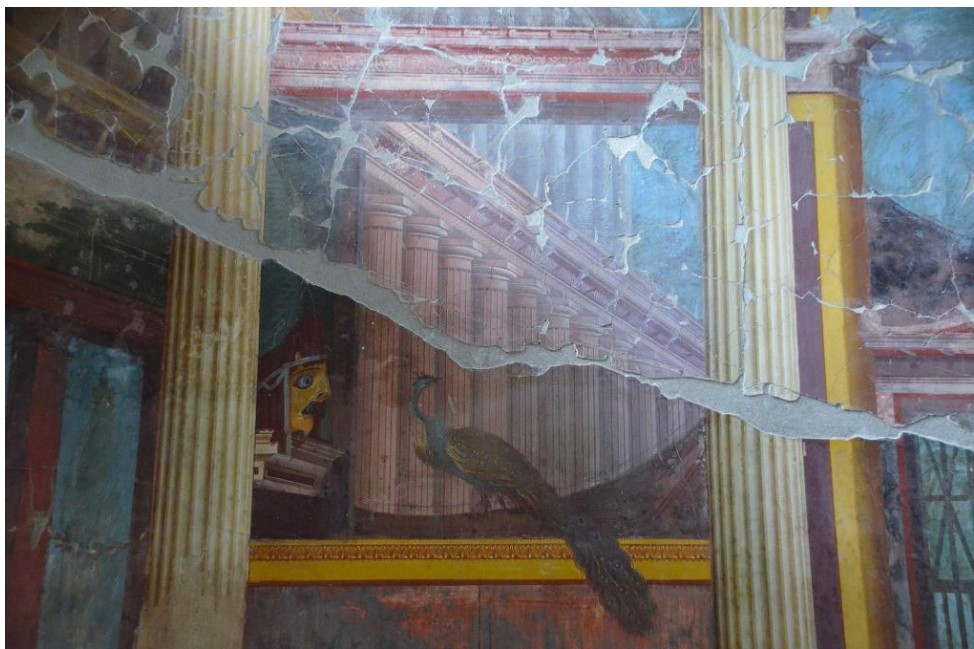

**Figure 3.** Villa Oplontis, Torre Annunziata (Naples), 1st century CE. Room 15, East Wall, detail. Photo: Carole Raddato CC-BY-SA 3.0.

Dionysius illustrates the aesthetic experience of Lysias' rhetoric as opposed to Isaeus' by comparison to the enjoyment of a gracile archaic line drawing versus the chiaroscuro and illusionism of a Hellenistic painting, and likens Isocrates' lofty style to the grand statues of Phidias and Polykleitos (*Isaeus* 4; *Isocrates* 3).[24] These comparisons are anachronistic, with artistic references ranging from the archaic to the Hellenistic periods used of orators who were contemporaries in the 5th and 4th centuries BCE. Such pictorial contrasts are reflected in first century BCE Roman painting, as at Cubiculum B of the Villa Farnesina, depicting a fictive pinacotheca where archaic Greek drawings flank an illusionist Hellenistic painting.

*Skiagraphia* (and the related *scaenographia*) concerns the presentation of things or images in such a way that their appearance belies their reality when seen from a certain viewpoint. Such presentation employs techniques such as relief, foreshortening and perspective, which exploit optical deceptions—what David Summers, with sophistic inflection, calls the "*fallacies* of sight" (Summers 1987, pp. 42–49). A perspectivist presentation concerns relativism insofar as it presupposes a certain vantage point and involves the acceptance of a misleading appearance in cognizance of its difference from reality, much as Gorgias speaks of the audience's "wisdom" in its voluntary acceptance of *apatē*.

In post-antique, Renaissance discussions of the painter's perspective, the scenographic illusion, understood and enjoyed, was referred to as the *piacevolissimo inganno*, the "most pleasing deceit," (Cini 1967, p. 93), although Renaissance literary criticism also offers rehearsals of the association of fallacious argument and optical trickery.[25] It is also important to note that while perspective entails relativism, it also generalizes the relationship between image and observer; any observer in a certain vantage point will have the same experience, as displayed in the various Renaissance schema for "legitimate construction." This relative yet generalized viewpoint is distinct from the beholder's imaginative freedom and activity as discussed, for example, by Philostratus.[26] It should be emphasized that perspective is discussed here as an illusionist technique, not as the projective geometry derived from Euclid via medieval optics which enabled the rationalist construction of pictorial space, albeit a rationalist construction anchored in viewpoint.

The legitimation of *apatē* requires the delimitation of a sphere of aesthetic experience, as we find in Augustine's characterization of art as *mendax*: true to its nature when it deceives with illusion (*Soliloquies* II.10–18; II.35). This characterization reflects a view of art as profoundly sophistic, yet in the dialogue paintings and sculptures, like the performances



of actors, are exonerated from deceit because they do not aim to mislead. It is to the sophists that we can trace the idea of a fictive or "poetic" truth, as when Philostratus opens the first book of the *Imagines* with the assertion that whoever scorns painting is unjust (*adikei*) to truth (*alētheia*) and to the wisdom bestowed on poets—language that seems to echo Gorgias' concept of the due deception of tragedy.[27] The *Imagines* proceeds to embroider the theme of artistic *apatē* with the claim that painting is more sophistic (*sophizetai*) than sculpture because it allows us to recognize the joyful, sorrowful or frenzied states of the soul (294) and the *ekphraseis* evoke the scents, music or words "arising" from the depictions as we enter into the pictorial illusion.[28] The voluntary participation in illusion is even more intensely marked in the *Ekphraseis* of Callistratus, in which a bronze statue is envisaged as softening into flesh at the touch of the viewer's hand (Ekphraseis 8; Onians 1980, p. 6). The most intricate example of *apatē* in the *Imagines* appropriately comes in the Narcissus ekphrasis, in which the conjunction of appearances made by nature and by art is exemplified by a bee resting on dewy flowers that may be a real bee deceived (*exapatētheisa*) by the image or a painted bee that deceives (*exēpatēsthai*) us. Here we have a regress of aesthetic trickery, with a Zeuxis-style *apatē* concerning a beguiled animal compounded with human viewers voluntarily "deceived" by an illusionist insect in an illusionist painting that depicts a man deceived by his reflection "painted" in a pool. The illusionist devising (*sophizein*) of painting that deludes animals and unprepared observers is a prerequisite for the sympathetic engagement that enables us to become wiser through our experience of art through our willing participation in the illusion.

Such sophistic-poetic "truth" would seem to be a realm of invented appearances associated with vision, enchantment or dream that powerfully affects the psyche and is at once illusory and divulgatory. Mario Untersteiner, in his important study, *I sofisti*, saw in Gorgias' work a recognition that in art idealism is inseparable from trickery (Untersteiner 1949, pp. 226–27). The enchanting effect of this realm of artifice is indeed powerfully conveyed in sophistic literature of various periods and in its critical portrayal in the Platonic dialogues: as Plato understood (and exemplified through his dual use of myth and dialectic), the crux of the matter lies in the character of this fictive "truth." We can thus see how much is at stake in sophistic conceptions of art—and how they could ultimately lead to the notion of a separate sphere of art, divorced from scientific verity and objectivity, whose revelatory character is indissolubly linked to its psychological perspectivism and whose "truth" may be merely a matter of artistic effect. One might contend that optical perspectivism, including techniques such as entasis, foreshortening and legitimate construction, which imply generalized conditions of vision from a particular viewpoint, has its more problematic counterpart in the psychological perspectivism that concerns the subjectivity of apprehension and reflective response.

## 4. Performative Responsiveness in Second Sophistic Rhetoric

With all this in hand, we can now turn to the accounts of imaginative freedom, invention and performative responsiveness in sophistic rhetoric, mindful of the question of the ends of such imaginative activity. I shall focus on Lucian's *De domo (The Hall)*, a fictive epideictic speech (or pair of *antilogoi*) concerning the description of an ornate hall, and two passages on image-making in Philostratus' *Apollonius*. The questions that arise in these texts provide an overview of the rhetorical presentation of sophistic aesthetics of the Second Sophistic, but they can be read as descendants of Gorgias' discussion of *logos* and its relation to vision and in light of Plato's critical analysis of the ontological status of images in *Sophist*.

Gorgias' *Helen* describes the shared power of words and sights alike to form or model (*typoō*) the soul (see Wilburn in this special issue). In *Helen* 15, Gorgias moves from persuasive *logos* as the cause of Helen's seduction by Paris to love, provoked by sight, and the remainder of the speech pursues the theme of the emotional impact of vision. If *logos* is a "great master" who "with a very subtle body accomplishes divine works" (*Helen* 8), the images (*eikones*) that the eye impresses or draws (*graphein*) on the mind affect us

even more forcibly, engendering overmastering states of love, great suffering and madness. While *logos* and sight are linked in their capacity to produce sorrow, pleasure and fear, only *logos* can banish negative emotions, error and ignorance and so annul injustice, unlike the powerful but less sophisticated compulsion of vision and enchantment.

This pairing or reflecting of words and images continues in later rhetoric as an element in speeches that adorn festivals, one of the principal civic uses of oratory during the Roman empire.[29] Aristides' *Panathenaic Oration* exemplifies such epideictic productions; the speech is offered at its end as an ornament to rival the embroidered *peplos* of Athena presented at the Great Panathenaea and includes a veritable pageant of psychagogic rhetoric as the imagined voyage towards Athens (*Panathenaic Or.* 10–12), which is figured as a procession, the spectacle of a dance or the stars encircling the moon, the purification of the soul for initiation in sacred rites, the revolutions of a choric dance, a joyful dream and the Homeric lifting of mist to disclose the gods (*Iliad* V.127). Such panegyrics present eloquence as a fitting response to the spectacle, such as Libanius' *Oration* XII on the Emperor Julian entering his fourth consulship, where Libanius declares that he will not be a dumb spectator before Julian. A counterexample appears in Ammianus Marcellinus' account of the triumphal entry of Constantius II into Rome in 357, where the imperial spectacle that transforms regalia into living things—banners with dragons that seem to hiss and writhe and actors who are immobilized like statues—is critiqued as a frigid spectacle that fails to elicit resonating words of praise (*Res gestae* XVI.10.6–8, 10).

This theme of eloquence as the proper response to visual splendor occupies Lucian's *De domo*, which opens with the speaker declaring that the hall's beauty impels him to compose speeches about it, filling it with his voice, in order to become part of its beauty and to make a return for the sight with speech. The hall is envisaged as filled with praise so that it echoes like a cavern (*De domo* 3), becoming like a resonating chamber for the praise it provokes. We move from an object that occasions a response to a kind of regress where the object echoes back the response that reflected the object; the speaker's response is in turn created via the *phantasiai* as the vision of the hall's beauty flows from his eyes into his soul, fashioning itself into the hall's likeness and sending out *logoi* (*De domo* 4). We might also recall that in *De sublimate* 15 *phantasia* signifies every form in the mind that gives birth to speech. In *De domo*, the idea that reflections and impressions are not just empty appearances but have affective agency thus reverberates through the speech, which indeed progresses to take for its theme visual scintillations.

This attribution of agency to visual impressions appears even in details, such as the praise of the gilded ceiling, which is not just ornate but illuminates the hall through its reflection (*De domo* 8). The theme reaches a crescendo with the argument that even animals delight in beauty, exemplified by the peacock (used by Dio of Prusa in *Oration* XII.2–3 as an image for the sophist), which spreads its tail in the springtime meadows, displaying its plumage to the sun and the flowers in an offering or challenge.[30] The passage culminates in a tour de force of sophistic rhetoric evoking the shimmering of the peacock's tail as its colors blend and shift hue with the movement of sun and shade, the plumage modified and readorned (*metakosmeō*) in the light (*De domo* 11).

The manifold, changing appearances of the peacock's tail are not just a feast for the eyes but an image for the beholder's activity in apprehending and generating the *phantasiai* that provoke *logoi*—the words whose brilliance and ornament will reflect the hall's beauties and add luster to them. The peacock passage, as a literal embodiment of purple prose or colored speech, is followed by an image that suggests the sense of longing beauty arouses; the hall moves us to speak as the prospect of the calm sea moves us with the urge to sail away from the shore (*De domo* 12). The image echoes the simile at the opening of *De domo*, when the urge to speak about the hall is likened to Alexander's desire to swim in a pure river: the longing to become a part of the beautiful carries the notion of immersion or oceanic abandonment. The theme of reflective brilliance, of *logoi* as a radiant mirror to *phantasiai*, has its counterpart or undertow in the idea of blissful dissolution in the beautiful.

The hint of compulsion in the oceanic simile leads into familiar images of enchantment and Siren song for the hall's beauty (*De domo* 13). At this point, another voice (or an *antilogos*) arises to shake off the bewitching spell and presents the counterargument that the hall's beauties are disadvantageous to an orator as theme and setting because the speech will fall short of visual charms. Splendid surroundings will overwhelm the speaker's skill and distract the audience as ornaments distract from a woman's beauty; less attractive surroundings by contrast form a foil for eloquence (*De domo* 15–17).[31] The second argument, however, discourages praise of the hall due to the very dominance of sight over language. If the Sirens could be escaped or disregarded, the Gorgons' beauty was irresistible, entering the vital parts of the viewers' soul and leaving them utterly changed and petrified in amazement (*De domo* 18). If words are winged, things seen remain and hold the viewer in their power (*De domo* 20).

If Lucian's *De domo* concerns the performative, sympathetic engagement with the beautiful, enabled by the *phantasia* and reflected through *logoi*, Philostratus' two discussions in *Apollonius of Tyana* (VI.19 and II.22) on mental images as the basis for artworks similarly exalt the imaginative powers of the viewer (Philostratus 2006). Philostratus' discussion at VI.19 appears to be informed by Dio of Prusa's *Oration* XII, which treats the human need to portray what is invisible and unportrayable with what is visible and portrayable and thus give the god a bodily form as a vessel (*aggeion*) for intelligence and *logos* with the force (*dynamis*) of a symbol (Dio Chrysostom 1932).[32] The god's body is, thus, not a likeness but the worthy and fitting container for a symbol, expressive of the qualities attributed to the god. *Apollonius* VI.19 can thus be read as a complement to *Oration* XII, since Dio had not indicated how the artist invents the form that will contain or convey the divine content.

In the passage, an Egyptian antagonist, Thespesiōn, responds to Apollonius' attack on the cult images of dumb (*alogos*) animals by questioning ironically whether Phidias and Praxiteles took impressions in heaven of the forms of the gods.[33] Apollonius replies that the fitting and beautiful images were molded by a "wiser demiurge" than mimesis, namely, *phantasia*, which creates things that it does not know, proposing (*hypothēsetai*) things with reference (*anaphora*) to being. The image of an animal, by contrast, may be admired for its likeness but diminishes the gods, as is also noted by Dio in *Oration* XII. (This derogatory view of mimesis also appears in the *Imagines* 308, where imitative details are dismissed as insignificant compared to the wisdom and *kairos* of the painting, and *Imagines* 294, where a modest definition of painting as based in mimesis and closest to nature is contrasted with a sophistic devising that claims it to be invented by the gods.[34]) In response to Thespesiōn's defense that the Egyptian images are venerable and suggestive (*hyponooumena*) symbols, Apollonius retorts that it would have been more suggestive for the Egyptians not to have used images in their temples and liturgy, so that worshippers could have imagined and delineated their own mental forms of the gods. Egyptian effigies lack beauty in their aspect and their suggestive power.

Thus, *phantasia* has a "wisdom" that consists in its invention of non-existent things, a point that recalls Gorgias' attested remarks in *On Nonbeing or On Nature* on imaginary creatures or events as illustrating our capacity to form conceptions of things that do not exist.[35] Apollonius therefore places images of the gods in the same sphere as the monstrous hybrids Scylla and the Chimaera or chariots crossing the sea, which are Gorgias' examples of non-existent, imaginary conceptions (pseudo-Aristotle, *Melissus, Xenophanes, and Gorgias* 980a12–13; Sextus Empiricus, *Adversus Mathematicos* VII.79–80).[36] As in Dio, the form of a deity's image is not a likeness, but in Philostratus we do not find the relationship of tangible vessel and intelligible content—unlike in Dio, who insists on the symbolic relation of the intellectual content and the figurative vessel, governed by the criterion of appropriateness and linked to its affective character. Instead, there is a movement towards relativism and subjectivism in Apollonius' proposal that it would have been more venerable and suggestive for the Egyptians to have left their holy places without images because the mind portrays and imagines (*anatypoutai*) things better than craft (*dēmiourgia*).

Imaginative exercise is not only the prerogative of gifted craftsmen but of anyone who frequents a temple and can replenish an aniconic space with the forms in their *phantasia*—here signifying imaginary forms rather than the appearances of things seen (*phantasmata*) that generate inner images. It is unclear how the plethora of mental images arising in each worshipper could meet the appropriateness (*proskēontos*—from *proskeimai*, to be involved in or attached to) that generally governs the making of cult statuary, exemplified by the "beautiful and reverent" Greek effigies, except in the sense that the mind grasps the attributes and qualities linked with the god whose form it invents.

Philostratus seems to pay little attention to the artwork as an existing object, seeing it as a more or less skillful materialization of an imagined form. If we might discern here a Platonic echo, whereby a painting is an apparent image of a phantasm, it must be juxtaposed with Philostratus' sophistic regard for the mind's creation of non-existent things it does not know—a formulation that recalls Gorgias but also evokes the conclusion of Plato's *Sophist* 268c, where the dialogue at last reaches the definition of the sophist as an ignorant practitioner of phantastic mimesis without knowledge of what he does. In short, Philostratus (or "Apollonius") appears to conflate (or confuse) the derogatory definition of the *Sophist* and the Gorgianic celebration of sophistic *logos* without considering the ontological character of nonbeing analyzed with subtlety by Plato and proclaimed in skeptical terms in *On Nonbeing*. *Apollonius* VI.19 does not address the source of the *phantasia*'s images; the form (as opposed to the attributes) selected for the cult image is connected with the meanings it carries only in the sense that it evokes awe and reverence in the beholder—that is, on the grounds of its appropriate affect.

The earlier discussion at *Apollonius* II.22 makes similarly sophistic claims for mimesis itself. In *Apollonius* II.22, Apollonius discourses to his disciple Damis in the temple of Taxila (Jandial, now in the Pakistani Punjab) while they contemplate silvered and gilded bronze relief plaques showing the *gesta* of Alexander and Porus.[37] Apollonius insists that mimetic art (*mimētikē*) is twofold, consisting of painting (*graphikē*), which employs mind and hand, and purely mental image-making (*eikazein*)—such as the distinction noted in VI.19. The latter is innate (*ek physeōs*) while the former requires art (*technē*), but the mimetic faculty is also essential to looking at artworks, since admiration of depicted things entails recognizing what is imaged: Philostratus speaks of having in mind (*enthymētheis*) what is pictured, using a term that evokes Aristotle's enthymeme or rhetorical syllogism, in which certain premises are unstated because they are self-evident.

Philostratus erodes the distinction between the imagination's role in judgement, when we use the image received from our senses to form concepts, and imaginative invention, when we "conjure" up a form in our heads (note our recourse to the sorcery analogy here).[38] The insistence on the two-fold character of mental image-making (including image reception via pre-existing conceptualizations) and material image-making exemplifies the sophistic concern with reception and affect whereby the received image is reflected back through the audience's response (notably through a *logos*) in a kind of regress of *phantasiai* that are invented, received and reconstituted.

The discussion in Philostratus' *Apollonius* significantly opens with images that are chimerical projections: centaurs and *tragelaphoi* ("stag-goats") in the clouds.[39] The naïve supposition that such forms are created by the gods is emended to the statement that such likenesses are created by chance but composed by humans—who are by nature imitative (*mimētikon*)—into pictures. Philostratus uses the *hapax legomenon* "*anarrhythmizein*" to describe the process by which we form such random shapes into projected images, where the *ana-* prefix to *rhythmizein* may suggest arranging or composing anew, or a composition wrought upon something. The centaurs and *tragelaphoi* of *Apollonius* II.22 are doubly non-existent, as chimaeras and as forms in the clouds; they show both the capacity to conceive unreal things and the ability to project likenesses on the basis of random features.[40] It is worth recalling here that the distinction between random and essential features, or properties, is itself a construct of Aristotle's logical categories.

If our ability to form representations is anchored in our image-making capacity, this capacity can perform acts of projection that see forms where none exist. This goes beyond our vulnerability to optical illusions due to medium or distance; it involves a willful reading of figures into the potentiality of random shapes. If Gorgias' *On Nonbeing* adduces imaginary hybrids as evidence for our capacity to imagine non-existent things in a skeptical ontological and epistemological argument, Philostratus simultaneously identifies the imagination as foundational to our knowledge of things and celebrates its deliberate distortions when it projects non-existent forms onto existing things. In this account, mimetic activity parallels the involuntary generation of forms from fortuitous resemblances ascribed to the love-sick and delirious, who project images onto such random shapes as stains (see Aristotle, *De insomnis* 460b11–14)—the activity that forms the analogy for the grotesque hybrids of Horace's *Ars poetica* 1–13. In Ficino's commentary on Plato's *Symposium* 203b, where Eros is called a sophist, the sophist as purveyor of the false as true has a counterpart in the lovers "blinded by the clouds of love" who accept false things for true things (Ficino 1998, p. 111). Again, we find the duality of proffering and reception of images.

The form made by chance has a long history In Western art and its literature, often standing as a figure for the artist's inventive fecundity, as in Leonardo da Vinci's advice to artists to see pictures in stains and in the fire as a kind of propaedeutic—a kind of artistic *kairos*.[41] In *Apollonius* II.22, forms made by chance, often associated with the "fevered" states of love or madness, exemplify our reading of images and thereby our conceptual activity—a more extensive claim than the artistic injunctions to project fantastical forms as an inventive exercise. The activity of the *mimētikon* thus seems like a kind of voluntary madness or delusion. We invent non-existent forms and then project such empty phantasms onto existing objects without a basis in the thing perceived. Philostratus' sophistic celebration of image-making is potentially far more damning of art's phantastic mimesis than Plato's description of pictures as a kind of dreaming in waking life in *Sophist* 266c. Behind Philostratus' account of chimeric image-making we can also glimpse another famous sophistic proposition discussed by Plato, namely, Protagoras' postulation of the subjectivity and relativity of experience (*Theaetetus* 152a–152d).

*Apollonius* II.22 recalls the most unsettling implications of the early sophists' skeptical and relativistic arguments. A theory of image-making as the basis of conceptualization that identifies it so closely with the manipulation of appearances and with bottomless subjective delusion does not provide a positive model of imaginative freedom. In *Helen* 11, Gorgias notes that if we all had memory of the past, knowledge of the present, and foreknowledge (*pronoia*) of the future, *logos* would be the same (*homios*). But since the past is unremembered, the present unexamined and the future unknown, most people possess only an opinion (*doxa*) as a symbol in the soul; such *doxa* extends its insecurity and inconstancy to its objects.[42] Philostratus' phantastic mimesis suggests the extreme point of such unexamined and inconstant projection: it effectively inverts Dio's model of art as a body created as an apt vessel for an intelligible, incorporeal content.

Before we turn to assess what kind of inheritance these sophistic ideas about images bequeathed to later ages, we can note, by way of summary, that even the few texts selected for comment reveal a range of highly suggestive themes. The most profound theme concerns the ontological status of imaginary phantasms and painted appearances, since behind the passages in *Apollonius*, one may discern the discussions in *Sophist* about the nonbeing of images and the relationship between divine and human appearance-making.

Another issue concerns the relationship between speech and images. While sophistic oratory is associated with verbal–visual analogies, as Cicero notes in *Orator*, it also envisages distinctions between the effects of images and *logos*.[43] We saw in Gorgias' *Helen* that speech and sight, associated with love, could both affect their audiences powerfully. Speech persuades and creates and expels fear, pity or sorrow, while sight can bring pleasure, the contentment of beauty judged according to norms, and the good born from justice, or, alternatively, fear that expels thought and brings suffering and folly. The visual–verbal analogy illustrates affective force, and, given the centrality of tragedy to Gorgias' discussion

of moving speech and its "due" deception, allusion to the relationship of words and spectacle is inescapable. While both excite emotions, only *logos* is said to expel them, and Gorgias ends *Helen* by claiming that his *logos* has banished the injustice and ignorance of *doxa* (*Helen* 21)—although this is undercut by the final assertion that the entire speech has been a plaything. The claim that poetic *apatē* is *dikaioteros* should also be read in the light of Gorgias' assertion that *logos* banishes injustice.

We might thus suggest that in sophistic-artistic analogies, art is illustrative of deceptive appearances (as in philosophical critiques), of intense or compulsive affect upon a viewer (described with allusions to enchantment, wonder, *goētia* or awe-struck animals), or as provoking reflective responses in which the writer's phantasies "echo" the stimulus of the object, conceived as an appearance created in and by the artist's *phantasia*. It is less clear whether artworks or spectacles are conceived as possessing the restorative movement described by Gorgias as heightened emotions are expelled or cathartically purified. If artworks cannot accomplish the movement from *doxa* to greater justice and wisdom, we have seen that they are at least occasions for celebrating imaginative response, as one appearance (*phantasia*), whether an artwork or a natural image (e.g., a dream or reflection, etc.) provokes a further response. The suggestive character of this kind of mirroring in which one imaginary image evokes another is, however, built on the lack of distinction between sensory impressions and invented images (what the Stoics differentiated as *phantaston* and *phantastikon* (Bundy 1927, p. 89)). This is most striking in *Apollonius* II.22, where the role of the *mimētikē* in concept-formation and in generating inventions of escalating subjectivity is affirmed without distinguishing the two operations. This creative ambiguity between impressions, invented images and mental forms is aesthetically powerful but conceptually weak.

Before we consign sophistic aesthetics to sub-philosophical literary reflection, we might briefly recall another area of sophistic-Platonic influence that is also significant for the Renaissance. This concerns the theurgic tendencies of post-Iamblichan Platonism. Its interpenetration with sophistry in the later empire is evident in Eunapius' *Lives of Philosophers*, written in the late fourth century, in which we find portraits of philosopher-sophists such as Priscus, who were proponents of the esoteric Neoplatonism venerated by Eunapius (Eunapius 2007).[44] By this period, sophists and philosophers were no longer antagonists but could be seen by non-Christian authors as protectors of the *paideia* encompassing literature, myth and religion that was threatened by Christian monastics and barbarians, an attitude exemplified by Julian's endorsement of *logoi* and *hiera* and his promotion of "thaumaturgic" philosophers such as Maximus of Ephesus (Eunapius, *Lives of Philosophers* 473–78).[45]

In the convergence of sophist and philosopher-hierophant as sacral custodians of *paideia*, the psychagogic character of sophistic *logos* as a kind of *goēteia* takes on another guise. The sophist as conjuror or sorcerer who purveys phantastic illusions like dreams acts within a cosmology in which natural illusions of the kind called divine appearance-making in *Sophist* 266b-c—shadows, reflections, dream images—are created by the daemonic beings that inhabit the analogically connected levels that comprise the structure of reality in (for example) Proclus' philosophy. Such Platonic links between daemons and sophistry were already suggested by the reference to Eros as a daemon and sophist in *Symposium* (203b), which recalls Gorgias' allusion in *Helen* 8 to *logos* as a "great master with a very subtle body who accomplishes divine works".

The notion of the demiurgic sophist indeed appears in the scholion on Plato's *Sophist* attributed to Proclus' circle, which was reproduced (assigned to Proclus) in Ficino's annotated translation of *Sophist*. Noting that Plato called Eros, Hades and Zeus sophists and citing Iamblichus' view that *Sophist* concerns the sublunar demiurge, the scholion calls the sophist many-headed.[46] This corresponds to the multiple levels of sophistry, encompassing Hades the sublunar demiurge (*Cratylus* 403a–404b) who imitates the heavenly artificer, Eros (see *Symposium* 203b) who presides over the reciprocal attractions and enchantments in nature, the philosopher as imitator of the demiurge and possessed by love, and finally the base sophist as imitator of the philosopher.[47]

In this Neoplatonic reading, which relates *Sophist* to *Timaeus*, everything beneath the celestial demiurge who fashions the intelligible species is a kind of sophistry that participates in some kind of imitation or conjoining of the existent and the non-existent, encompassing the changing, varied forms of generated things with their "imaginary essence," shadowy and deceptive daemonic productions, and human imitations, whether these follow the "higher" sophists (Hades or Eros) or the deceptive verbal magic of the maligned sophist.[48] The Neoplatonic reading places the world of nature and of human creation at the level of sophistry, indicating that the discussion of sophistry and art was not confined to select themes such as illusionist techniques but could underpin the conception of art in its totality. We might recall that Proclus defended mythic poetry—condemned for its sophistic mimesis in *Republic*—as daemonic because it resembles the revelations through symbols brought to us by daemons in dreams.[49] In an ensouled cosmos structured through correspondences that may all be activated by theurgy, the "magic" of art's illusion or fictive truth is just one level of likeness and imaginary being and may be deployed to channel forms or essences at higher levels. This may provide an ontological model in which the imaginative resonance of sophistic aesthetics is salvaged from relativism and subjectivity, but what is lost in this deterministic cosmology is *kairos*, embedded in the potentialities of the moment or chance.

## 5. Sophistry and Renaissance Art

I noted above that the habitual denunciation of sophistry obscures perception of the degree to which post-antique art—and writings on art—engaged with the practices and conceptions that were in antiquity considered sophistic. Much of the ancient art rediscovered during the Renaissance was Roman imperial sculpture copied from earlier Greek originals—statuary created to adorn the monuments and public spaces, such as imperial thermae or gymnasia where the "deutero-sophists" performed or which they erected, such as the *Farnese Bull*, recovered from the Baths of Caracalla in 1546 for display in Palazzo Farnese (see Thomas 2017, pp. 181–201)[50] (Figure 4).

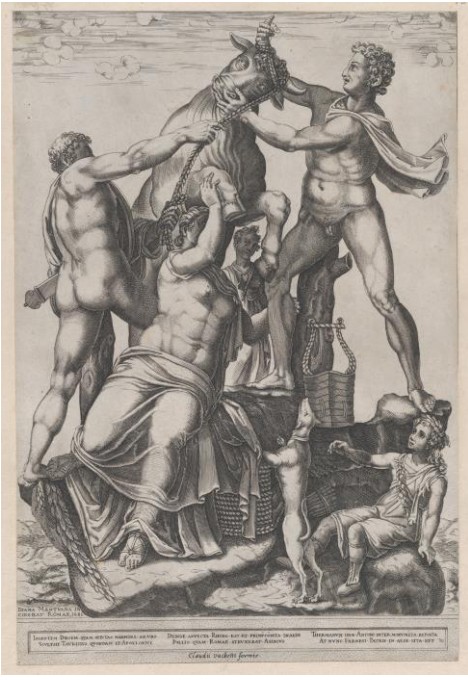

**Figure 4.** Diana Scultori, *Speculum Romanae Magnificentiae*, 1581, *Farnese Bull*. Metropolitan Museum, Harris Brisbane Dick Fund, 1941.

When Pliny's accounts of the contests in illusionist *apatē* between Zeuxis and Parrhasius were read and imitated, as in Filarete's and Giorgio Vasari's accounts of Giotto deceiving his master Cimabue with painted flies, we should consider the attitude toward

illusion as well as the display of literary allusion (antique and Humanist, notably Alberti's imitation of Lucian) contained in the reference (Rawlings 2008, pp. 7–13). When Alberti adduces Protagoras' dictum on man as the measure of all things in *De pictura* and calls Narcissus the inventor of painting, extolling art's power to make the dead appear living, we are confronting an indisputably sophistic view of art, complete with hinted *goēteia* in the necromantic suggestion of the dead revivified by art (Alberti 1972, pp. 52–54, 63–64, 121).[51]

There are three particular areas in Renaissance art where attention to a sophistic inheritance would be enriching and merit further exploration. The first concerns illusion, or better, illusionism, described by Sven Sandström in peculiarly sophistic terms whereby "the picture becomes the mirror in which the human being can gaze on the god with impunity" (Sandström 1963, p. 186). The second is the vogue for grotesque ornament, inspired by the rediscovery of the Neronian Domus Aurea as well as Roman (particularly Trajanic and Hadrianic) "peopled acanthus" relief carving. Such ornament was overtly sophistic in its presentation of "things that are not and cannot be," as Vitruvius complained in *De architectura* VII.5—a passage glossed by Daniele Barbaro with reference to the dream images brought by the *phantasia* in sleep and as the pictorial equivalent to the sophist who creates "monstrosities" such as our phantasy represents (Barbaro 1567, p. 321). It has been argued that the Renaissance *grottesche* become a kind of motif for phantastic invention, appearing in treatise literature where the eikastic–phantastic distinction of *Sophist* arises, and associated with figures projected onto chance forms (Guest 2019, pp. 155–200).[52] The stereotyped *bizzarrie* of the *grottesche*, in which imaginative exhaustion itself becomes a theme, can be viewed in terms of their role as figures that allude to certain ideas about imaginative invention (Figure 5). If the *grottesche* reflect the pervasiveness of topical invention in later Renaissance art, their displays of metamorphoses founded in chance resemblances exhibit that relationship between topics and *kairos* we noted in Renaissance recoveries of sophistic literature.

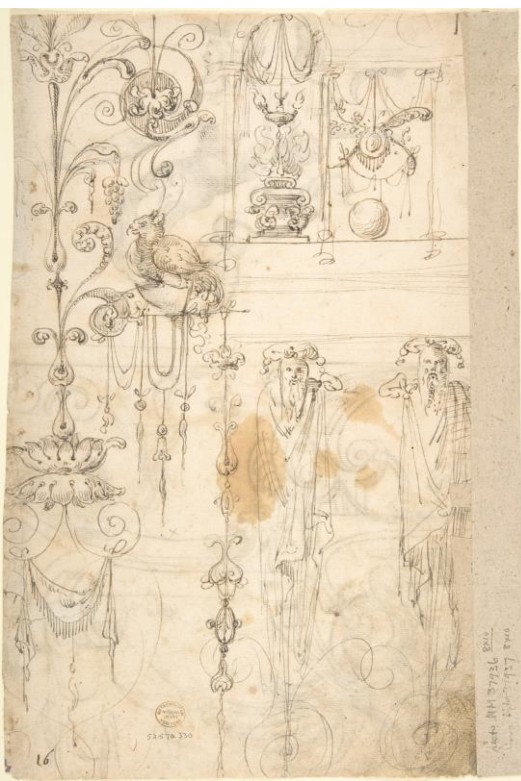

**Figure 5.** Andrés de Melgar attrib. Study with *grottesche*, ca. 1545–60. Metropolitan Museum, The Elisha Whittelsey Collection, The Elisha Whittelsey Fund, 1952.

The third area has its roots both in sophistic writings about artworks and the interest of Hippias in *archaiologia* (Plato, *Greater Hippias* 285d).[53] This concerns the antiquarian aesthetic that privileges the fragmentary state of artworks, provoking phantasies of restitution in the viewer's imagination. Renaissance antiquarianism also sustained the notion of *goēteia*, given the pervasive necromantic language used of recovering antiquities as a return from Hades.[54] The antique fragment can be seen as a peculiarly privileged case of an artwork whose significance lies in its reception, with the subjectivity and variability this entails (Figure 6). Scholars of Renaissance antiquarianism have noted the vulnerability of antiquities in relation to their audience and the tension between the fragmentary object and the imagined completion in the viewer's mind that "points to a greater wholeness than would any complete works" (Barkan 1999, pp. 124, 207). As Salvatore Settis observes, antiquarianism involved values and relationships quite different from the insistence on perfection and integrity in Aristotelian-Thomist aesthetics, where completion enables structural analysis and forms a prerequisite for *claritas* (Settis 1985, pp. 375–486). The antique fragment instead inspires a kind of improvisational completion in the viewer's imagination—*kairos* enters into the relation of viewer and object.

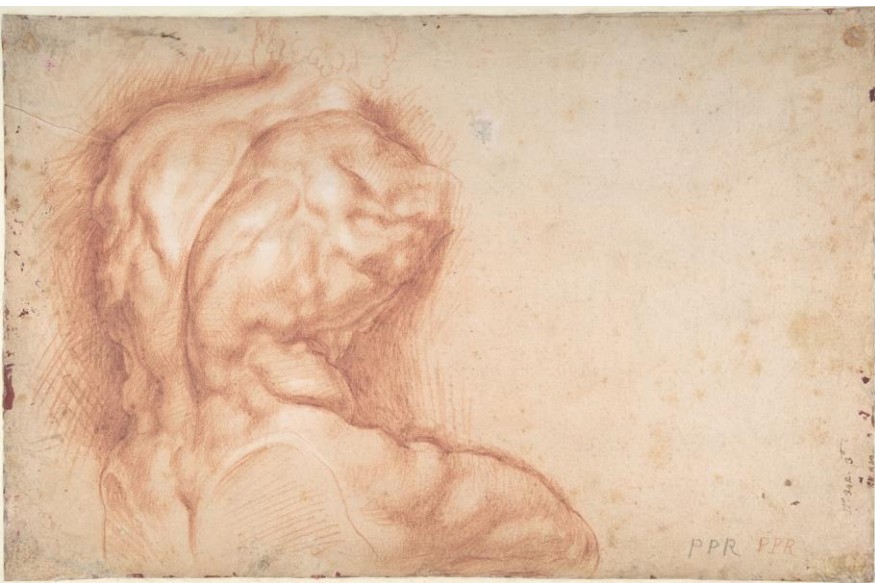

**Figure 6.** Rubens, drawing of Belvedere Torso, red chalk, 1601–02. Metropolitan Museum, Purchase, 2001 Benefit Fund, 2002.

These themes appear in seminal documents of Renaissance antiquarianism, such as the letters composed by Manuel Chrysoloras to Demetrius Chrysoloras and his epistle, *On the Old and New Rome* (1411), in which the Byzantine scholar describes Roman ruins (Niutta 2001; Chrysoloras 1886, *PG* CLVI, pp. 23–54, 57–60).[55] They show Chrysoloras working in the tradition of *paideia* launched by the sophists as he responds to the imperial ruins via citations of Libanius (see Smith 1992, pp. 136–42). Chrysoloras elevates the effigies of things over their models, noting our admiration for sculpted details such as a leaf or a vein that we barely notice in the living originals: he would prefer a ruined fragment of a horse by Praxiteles or Phidias to the perfect living animal. Confronting aesthetic norms concerning structural integrity, he proclaims that beauty appears not only in composite and unified things but also in things divided (*diairethenta*). In Aleardi's 1454 Latin version of *On the Old and New Rome*, Chrysoloras' contrast between effigies and originals is more forcefully expressed as "*Nec composita et constructa solum, sed disiecta et quoque et diruta*"—broken up and pulled asunder (Niutta 2001, p. 40; Chrysoloras PG CLVI, 25).

For Chrysoloras, the vision and presence (*autopsia . . . kai parousia* [*PG* CLVI: 28]) that he admires in ancient carvings that seem to live co-exists with the beauty of the fragment.

The thread joining the two is the viewer's imaginative engagement that makes ancient stones appear to be living men who transport the spectator into direct vision of the past and invests the onlooker in the pathos of the mutilated statue or in its projected restitution. The "greater wholeness" perceived by Barkan in conjectured perfection concerns the work of the imagination that completes or replenishes the object, or rather the object–viewer relation. Barkan's intuition is, thus, in direct line with the sophistic conceptions of art I have sketched out here.

I opened this essay with a question about how we can trace the influence of sophistry on visual art, especially on Renaissance art, in the face of the numerous condemnations of the sophists inherited from antiquity and rehearsed by the Humanists. As we look more carefully, we can discern that preoccupation with the transformative imaginative reception of artworks in the viewer's mind did indeed feature in Renaissance literature on art, becoming thematized in cases such as the *grottesche* and the ruin aesthetic where the viewer completes or composes in phantasy an imaginary object whose notional wholeness stands in creative tension to the actual artefact. While the space of this essay only permits a tracing of themes, I would like to conclude by signaling a need for the larger work of tracing lines of transmission, exemplified by a figure such as Chrysoloras, who represents a direct channel from the sophistic rhetoric of late antiquity to Humanist Florence with its "discovery" of perspective and cultivation of figural *enargeia*.

If this essay has suggested how we might start to cultivate a narrative that recognizes the presence of sophistic themes in pre-modern European art, we might also reflect on the post-Renaissance and modern aesthetic conceptions that are consonant with the sophists' exploration of the artwork as constituted through the viewer's imaginative response, founded in turn on the phenomenal character of experience. We can certainly find many examples in which subjectivism, voluntary deception, or the notion of art as stimulating internal reflection and engagement with appearances, reappear in modern art. Among such cases, we might adduce the reappearance of *apatē* in Samuel Taylor Coleridge's "willing suspension of belief" (*Biographia Literaria* XIV), Walter Pater's invocation of the ecstatic flood of impressions in the conclusion of *The Renaissance*, Sigmund Freud's exploration of the landscape with ruins as an analogy for mental life (as in his description of Rome as a "psychical entity" in *Civilization and its Discontents*), E.H. Gombrich's enlisting of Philostratus as the first theorist of projective psychology in *Art and Illusion*, and contemporary artistic experiments in perceptual art, such as the oeuvre of James Turrell, dedicated to engaging viewers with their own seeing, and Brion Gysin's *Dreamachine* (1959, reprised 2022), where flashing lights shone on closed eyes stimulate perceptions, images and emotional states that vary with each spectator. The common thread in these disparate cases is the focus on the experience of perception rather than the qualities of the artwork (as in the Thomist criteria of beauty, for example). I would like to end this essay in establishing the place of sophistic writings on art within Western aesthetics by suggesting that these writings constitute the first attempt in the European context to articulate the nature and significance of art from the perspective of the reflective viewer.

**Funding:** This research received no external funding.

**Institutional Review Board Statement:** Not applicable.

**Informed Consent Statement:** Not applicable.

**Data Availability Statement:** No new data were created or analyzed in this study. Data sharing is not applicable to this article.

**Conflicts of Interest:** The author declares no conflict of interest.

## Notes

1     Prior to the Renaissance, medieval allusions to the sophistic character of artistic illusion appear repeatedly in Alan of Lille's
      *Anticlaudianus* (ca. 1180) and *De planctu naturae* (ca. 1160s); see (Guest 2016, p. 61).

2     The term "Second Sophistic" was coined by Philostratus in *Lives of the Sophists* to describe the practitioners and teachers of Greek
      rhetoric and declamation in the Roman Empire, flourishing under philhellene Antonine emperors and exemplified by such
      second century figures as Herodes, Polemon and Scopelian. Philostratus distinguished this "second" Sophistic, whose progenitor
      he said to be the fourth century BCE Attic orator Aeschines, from the sophists of the classical period such as Gorgias, Prodicus
      and Protagoras, whom he describes as philosophically-engaged rhetoricians. Philostratus' distinction is loose: extemporaneous
      declamation—a key activity of Second Sophistic—originated with Gorgias, and Philostratus leaves a gap of centuries between
      Aeschines and the reappearance of sophists in Second Sophistic with Nicetes of Smyrna in the first century CE. As the Second
      Sophistic has grown in scholarly interest the range of writers and productions associated with it has expanded until the term has
      become almost co-terminus with Greek (and some Latin) literary culture in the Roman Empire (for the issues of definition, see
      (Korenjak 2018)).

3     MacPhail (2011, pp. 39–40) notes that "the extant speeches and major fragments of the sophists, except for those conserved
      in papyrus, had all reached Italy by 1492." Aldus' *Oratores Graeci* (1513) provided a printed edition. Mack (1993, pp. 90–91)
      notes Valla's contempt for Aristotle's *Sophistical Refutations* in Valla's attempts to create a non-scholastic, rhetorical dialectic. On
      Protagoras, see Trinkaus (1983) (Trinkaus notes that his essay was written in 1971).

4     Cassin (2017, pp. 143–56) argues that rhetoric emerges with a Platonic and Aristotelian project to tame *logos* by shifting from
      the temporalities of *kairos* to the spatialization of speech, in which the *topoi*, the "places of crafted speech," play a key role. On
      Agricola's association of the topics with extemporaneous oratory (associated with *kairos*), see (van der Poel 2019).

5     Against the sophists, see *Strōmateis* I.3, I.8. The *Strōmateis* was admired by Humanist cultivators of *varietas* and encyclopaedism
      such as Poliziano, who praises it in the dedication to his *Miscellanea* I alongside other Second Sophistic miscellanies such as
      Aelian's *Poikilē Historia*; see Poliziano (1498, fols. A$^1$r–A$^1$v).

6     Summers (1979) discusses the sophistic elements in his study of *phantasia* in art theory pertinent to Michelangelo, but they are
      subsumed under Aristotelian, Platonic and Augustinian conceptions.

7     The passage in *De divinatione* on forms immanent in matter follows Cicero's derision of images created by chance (I.13.23). In
      *Metaphysics* (Γ 1002a), Aristotle speaks of every kind of shape as equally present in the solid.

8     The Stoic *phantasia kataleptikē* or "cognitive impression" constituted a secure and accurate perception of its object that could
      be a criterion of truth; see Cicero, *Academica* II.77–78; Eusebius, *Praeparatio Evangelica* 14.6.12–13; Sextus Empiricus, *Adversus
      Mathematicos* VII 241–52; (Lories 2003, pp. 47–77).

9     Plato uses Proteus as an image for the sophist (or rhapsode) who evades dialogic questioning; see *Euthydemus* 288b–c, *Euthyphro*
      15d, *Ion* 541e; (McCabe 2008, pp. 109–23).

10    The *genus pictum* is the smooth and flowing middle style, characterized by its charm and use of figures of thought and speech,
      with which it is "painted," which arose with the sophists but was also used by philosophers (e.g., Xenophon, Plato and Demetrius
      of Phaleron).

11    The Philostratus (ca. 170–247/250 CE), who authored the *Lives of the Sophists* and *Apollonius of Tyana*, is said by the *Suda* to have
      been the great-uncle and father-in-law of the Elder Philostratus, who is thought to have composed the first series of the *Imagines*,
      while his presumed grandson (Philostratus the Younger) penned the second series. See Flinterman (1995, pp. 5–14), who cites
      Schmid's hypothesis that *Imagines* I was composed by the Philostratus who authored *Apollonius* and the *Lives*, with the Younger
      Philostratus being his grandson.

12    Cicero's *Academica* (II.7.22, II.23.23–26.84) rehearses skeptical and Stoic contrasts concerning *phantasiai*. The contrast with Epictetus'
      *Enchiridion* 1 on control over the *phantasiai* is emphatic. MacPhail (2011, pp. 28–29) notes that the genealogy of skepticism in
      Plutarch's *Table Talk* (652B) places Protagoras at the head of the skeptical tradition.

13    Cicero (*Orator* 37–42) discusses epideictic as proper to sophists.

14    Photius (1958, p. 187); Michael of Thessalonika, quoted in Mango and Parker (1960, p. 237).

15    On the dynamic description of perception and the associations of marbles and seas in natural philosophy, literature and theology,
      see (Onians 1980; Gage 1993, p. 57; Barry 2007).

16    On the *topos* of striated marble revetment as "flowery meadows," see Maguire (1998); Justin Willson. A Meadow that Lifts the
      Soul: Originality as Anthologizing in the Byzantine Church Interior. *Journal of the History of Ideas* 81: 1–21. Examples include *Greek
      Anthology* I.10 on the narthex of the Church of Polyeuctus, decorated by Juliana; Procopius on the marble revetment of Hagia
      Sophia (*On Buildings* I.1); Leo VI, *Sermon* 34; Paul the Silentiary, *Descriptio S. Sophiae*; Joannes Geometra (tenth century) on the
      marbles of Hagia Sophia as flowers that will never wilt (*PL* 106, 943); Philagathos (twelfth century) on the Cappella Palatina,
      Palermo.

17    Gorgias, *Helen* 10, 14. On the magic-*logoi* analogy, see (de Romilly 1975; Swist 2017, pp. 431–53).

18    See (Gutzwiller 2010, pp. 337–65, esp. 340–42); Dionysius of Halicarnassus *Lysias* 14.1; Strabo, I.1.10 on poets as aiming
      at *psychagōgia*.

[19] See (Garzya 1987, pp. 149–65) on *diakioteros* as signifying what is right or due.

[20] Plutarch, *De gloria Atheniensis* 348c, reports Gorgias' description of tragedy as *thaumaston*. Simonides also devised the aphorism on painting as silent poetry and poetry as speaking painting, transmitted by Plutarch in *De gloria Atheniensis* 347a and *De audiendis poetas* 18a without reference to Simonides.

[21] The term "Third Sophistic" emerged with Pernot (1993, p. 14, n. 9) designating fourth century pagan sophists; other scholars adopt different chronologies. Fowler (2014, p. 7) sees "Third Sophistic" as ending with the closure of the Academy in Athens in 529 CE and the last Platonists such as Olympiodorus (d. 570 CE); for Kaldellis (2009) it is identified with Michael Psellos (1017/18–ca.1078) and his heirs. See (Pernot 2000, p. 271; Amato 2006; Quiroga 2007, pp. 31–42; Fowler 2014, pp. 1–31).

[22] The other comparators for sophistic argument are the appearance of health and beauty created through dress and the appearance of gold and silver in base metals (*Sophistical Refutations* 164a26–164b25). At 169b22, Aristotle refers to sophistical reasoning and refutation as apparent but unreal (*phainomenon . . . mē onta*).

[23] Contrast the history of art summarized by Quintilian in *Instituto oratoria* XII.10, which illustrates stylistic development in oratory through that in art.

[24] Lysias was the exemplar of the "pure" and chastened Attic style; see Cicero, *Orator* 23–30.

[25] Passages in which optical illusion forms an analogy for specious argument appear in Annibale Caro's *Apologia degli Academici di Banchi di Roma contra Messer Ludovico Castelvestro* (1555) and Battista Guarini's *Il Verato* secondo (1593), his defence of the *Pastor Fido* against the criticisms of Giason Denore; see (Guest 2016, p. 574).

[26] Hills (1987, p. 18) criticizes Alberti's model of perspective as standardizing and impoverishing the beholder's relationship to the image compared to the sensitivity to movement of surface light in the gold grounds of Byzantine and Medieval mosaic and painting.

[27] On Philostratus' rehabilitation of Gorgias, see (Norden 1986, pp. 390–91).

[28] Thus, we "smell" the fragrance of the roses or apples (*Imagines* 298k, 301k) and "catch" the blood of the dying Menoeceus (I.4); at I.31 ("Xenia") the ruddiness of fruits is said to come from "within." Onians (1980, p. 5) notes that the descriptions of the younger Philostratus "read more into the painting than can ever have been visible."

[29] Aelius Aristides, in *Oration* III (672), part of his *Reply to Plato in Defence of the Four*, lists the civic duties abandoned by philosophers that correspond to the sophists' activities: honouring the gods, adorning festivals, advising citizens, comforting the distressed, settling civic discord and educating the young.

[30] Themistius (*Orationes* 336c) notes that sophists used bird analogies like rouge in their speeches. The notion that art strikes even beasts with wonder also appears in Dio's *Oration* XII.

[31] Women who are more beautiful with the most restrained adornment (*non nonnullae inornatae*) appears as a simile for plain style in Cicero, *Orator* 78.

[32] Dio notes that only the intelligent can revere the divine through contemplation of the heavens; others need representations that can be approached and grasped, like children who need to see their distant parents (*Oration* XII.60).

[33] The passage is preceded by Apollonius' defence of the role of ornament and decorum in religion to Thespesiōn as Elder of the *gymnoi*, whose asceticism Thespesiōn extolls by referencing Prodicus' Choice of Hercules.

[34] *Kairos* here probably means "time" or "state of affairs" as it was used in rhetorical manuals; see (Webb 2009, pp. 56, 61) *et passim*.

[35] The work is paraphrased by pseudo-Aristotle in *On Melissus, Xenophanes and Gorgias* (*MXG*) and by Sextus Empiricus, *Adversus Mathematicos* VII.65 and cited in Isocrates' *Helen*. See (Guthrie 1971, pp. 192–99; Wardy 1996, pp. 14–24; McComiskey 2002; Consigny 2001; Gorgias 2013).

[36] Untersteiner (1949, p. 267) suggests that chariots crossing the sea evoked the Oceanides of Aeschylus' *Prometheus Bound* (129–130).

[37] The bronze *pinakes* are said to be decorated with gold, silver, orichalcum and black bronze. Taxila appears in the Buddhist Jatakas as the capital of Gandhara and was an important centre for Buddhist learning. It was surrendered to Alexander and subsequently controlled by the Mauryan empire, the Graeco-Bactrian kingdom and the Kushan empire.

[38] The Stoics distinguished between the *phantasia*, the impression based on a real object, and the *phantasma*, the mental or dream image (see Diogenes Laertes, *Vitae philosophorum* VII.50).

[39] Centaurs and *tragelaphoi* are repeated figures for non-existent images or fantastical hybrids. See Aristotle's *De insomnis* 461b20–21; Plato, *Republic* 488a; Aristotle, *De interpretatione* 16a16–18.

[40] For comment on the passage, see (Onians 1980; Janson 1961; Gombrich 1960, pp. 154–69).

[41] Images projected onto chance forms are discussed throughout Renaissance art treatises: in Cennino Cennini; Alberti's *De Pictura* and *De statua* (hippocentaurs "painted" by nature in marble, chance forms as the origin of sculpture); Leonardo's *Treatise on Painting*; Doni's *Disegno* (1549); Armenini's *De' veri precetti della pittura* (1586); and Paulo Pino's *Dialogo di pittura* (1548), in which the forms "painted" by nature on tree trunks and in smoke are compared to reflections in a mirror. For references, see (Guest 2016, pp. 501–10; Janson 1961, pp. 501–509).

[42] The *symbolos* here may denote the symbol as a two-part token whose pieces are the *doxa* and the *psyche*.

[43] Cicero (*Orator* 65) said that the sophists used far-fetched metaphor as painters lay on varied colours.

44　　Synesius is a further case of a fourth century Neoplatonist and rhetorician; see (Munarini 2019).

45　　Eunapius' work is not doxographic but presents the philosopher as a thaumaturgic and divine figure, somewhat on the model of Christian hagiography.

46　　Cf. *Republic* 596d, where Glaucon calls the demiurge who makes natural things and the prototypes of man-made objects a sophist. On the scholion and Ficino's commentary, see (Allen 1989). The multi-headed sophist may derive also from *Republic* 588b–591d, where Socrates fashions a verbal picture of the soul as a varied beast (*thērion poikilon*), like Chimera, Scylla or Cerberus, with a ring of many heads contained within the image of a man.

47　　The Greek scholion speaks of attractions and repulsion uses in nature; Ficino translates this with *illecebras*, corresponding to his ideas of natural magic.

48　　Following Allen (1989), *Icastes*, pp. 95–112.

49　　Proclus, *On what Plato says in the Republic against Poetry* 85.26–86.23, in (Coulter 1976, p. 50). Proclus' distinction between eikonic and symbolic (daemonic) representations forms the basis for pseudo-Dionysius' symbolic theology, founded on the unlikeness between compared objects.

50　　On Second Sophistic funeral orations and mythic relief on sarcophagi, see (Ewald 2011, pp. 261–307). For the Second Sophistic as an artistic movement, see (Elsner 1998, pp. 4–8, 169–97).

51　　Alberti's application of Protagoras' dictum to the optical and perspectival issues of painting merits a study in itself; see (Trinkaus 1983, p. 174).

52　　See Guest 2019, pp. 178–79, for discussion of the phantastic–eikastic distinction in Comanini, *Il Figino* (1591) and Jacopo Mazzoni, *Difesa di Dante* (1587).

53　　Momigliano (1950, pp. 285–315) contrasts the history of institutions with political history as practiced by Thucydides, focused on individual and collective behaviour.

54　　Ciriaco of Ancona describes antiquarianism as the "divine art" that brings noble things back from Hades (see Brown 1996, p. 306, n. 50).

55　　Niutta (2001, p. 13) discusses the dedicatee of *On the Old and New Rome* as Manuel II Paleologus. Bruni's *Laudatio Florentinae Urbis* (1403–4) draws on Aristides' *Panathenaic Oration*, transmitted via Chrysoloras.

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

Zuccari, Federico. 1768. *Idea L'Idea de' pittori, scultori e architetti*. Rome: Marco Pagliarin.

