# Peer review of "Ut sophistes pictor: An Introduction to the Sophistic Contribution to Aesthetics"

_humanities, doi:10.3390/h12040058_

Round 1

Reviewer 1 Report

I love this essay.  I have some small concerns.  

1.  I like pieces that are a little more straightforward -- I like an introduction that signposts (a) what I already know, (b) what isn't really known yet, (c) and the roadmap by which the paper will introduce to me what I don't know yet.  I think this paper does those things, but not in the order I expect, and it's a little disorienting.

2.  There is a linguistic play in this paper -- a sophistication that I admire, but that also makes me wince and struggle a bit.  Example:  A moment when you use a french word to express what a Greek word is doing;  it's clever, but it's more clever than clear, to me.  Example:  A moment when you call your work an essai, as if I didn't know that English essay is derived from French essai, both of which basically mean a "trial."  It's rich, it's impressive, but it makes me work a little harder than I want to.  I feel similarly about some really, really, long sentences -- but recognize, I'm a rhetorician who teaches Writing in American universities, so I know I'm used to shortening everything.

Notes on page 1 attached.

Author Response

Thanks for your comments, I have found them quite helpful and have responded to the minor organisational issues, signposting, diction and overly long sentences. The text has therefore be revised in the light of these recommendations. 

Reviewer 2 Report

Reader Report for Ut sophistes pictor: An Introduction to the Sophistic Contribution to Aesthetics

This ambitious essay examines the potential contributions of Sophistic philosophy and thought to aesthetics in both the ancient and early modern world. All in all, it is impressively researched and argued, shows an enormous breadth and depth of learning, and quite deftly pulls together a number of distinct threads. It should quite clearly be published as part of this special issue, as it demonstrates both a novel contribution to the overall theme and an impressive level of scholarship.

I do, however, have a few small recommendations as the author moves forward in preparing their final submission. Despite the overall skill with which the author weaves these threads together, however, I still found myself a bit overwhelmed as a reader. Upon reflection, my sense is that this feeling stems from how intensely synthetic this essay is in the majority of its sections (notably 2-4).

In response to this feeling, I have one small piece of reader-response driven feedback for the author to consider as they move forward:

In sections 2-4 especially, would it be possible to support some of the claims tied to primary sources (e.g. examples of Byzantine ekphrastic literature in lines 166-187; discussions of Sophist and Republic in lines 230-243) with more (and more substantive) direct reference to texts? The tendency toward highly targeted quoting—often of a single word—can sometimes obscure important context that might assist the reader better following the author’s argument. I would point to the author’s quotation and reference of Gorgias’ Helen as an example of the style of quotation/reference that I think would improve the overall reading experience of the text.

To conclude, this is an excellent and ambitious essay. I learned much from the author’s wide-ranging inquiry and creative approach. But as a reader, I would personally benefit from a more even balance between analytic and synthetic approaches to argument and integration of primary source material.

Author Response

Thanks for your comments, I have found them quite helpful and have responded to the issues concerning quotation in Sections 2-4. I have thus expanded the discussions and provided more detail (and quotation), but I was also constrained by limitations of length from expanding too much.  The text has therefore be revised in the light of these recommendations.